# Genome Analysis of the G6P6 Genotype of Porcine Group C Rotavirus in China

**DOI:** 10.3390/ani12212951

**Published:** 2022-10-27

**Authors:** Ruixue Jiao, Zhaoyang Ji, Xiaoyuan Zhu, Hongyan Shi, Jianfei Chen, Da Shi, Jianbo Liu, Zhaoyang Jing, Jiyu Zhang, Liaoyuan Zhang, Shufeng Feng, Xin Zhang, Li Feng

**Affiliations:** State Key Laboratory of Veterinary Biotechnology, Harbin Veterinary Research Institute, Chinese Academy of Agricultural Sciences, Harbin 150069, China

**Keywords:** emerging, co-infection, porcine group C rotavirus, recombination

## Abstract

**Simple Summary:**

In this study, a case of infection with RVC was identified and the full-length genome of RVC in China was sequenced. A natural recombination event was observed in the RVC genome. Collectively, our data suggest that the RVC exists in China and it can be mixed with other diarrhea viruses, which brings new challenges to the prevention and control of diarrhea disease in pigs. Our data provide a new insight into the evolution of the rotavirus.

**Abstract:**

Swine enteric disease is the predominant cause of morbidity and mortality, and viral species involved in swine enteric disease include rotaviruses and coronaviruses, among others. Awareness of the circulating porcine rotavirus group C (PoRVC) in pig herds is critical to evaluate the potential impact of infection. At present, due to the lack of disease awareness and molecular diagnostic means, the research on RVC infection in China is not well-studied. In this study, diarrhea samples collected from pig farms were detected positive for RVC by PCR, and the full-length RVC was not previously reported for Chinese pig farms. This rotavirus strain was designated as RVC/Pig/CHN/JS02/2018/G6P6. A natural recombination event was observed with breakpoints at nucleotides (nt) 2509 to 2748 of the VP2 gene. Phylogenetic analysis based on nsp1 revealed that a new branch A10 formed. Collectively, our data suggest a potentially novel gene recombination event of RVC in the VP2 gene. These findings provide a new insight into the evolution of the rotavirus.

## 1. Introduction

Rotavirus (RV) is one of the main causes for diarrhea disease in both humans and animals [1]. It is a non-enveloped double stranded RNA virus and belongs to the family *Reoviridae*, genus *Rotavirus*. Its genome consists of 11 fragments encoding six structural proteins VP1-VP4, VP6, and VP7 and five or six nonstructural proteins NSP1 to NSP5 or NSP6 [2]. According to VP6, RVs can be divided into eight phylogenetic species, RVA to RVH [3]. Recently, two putative new RV species I [4] and J [5] were identified. RVC was first found in pigs in 1980 [6]. PoRVC is a global diarrhea disease [7]. The prevalence of PoRVC in suckling piglets is increasing and the emergence of new PoRVC strains with different genetic characteristics has caused economic losses for the pig industry, resulting in widespread concern about PoRVC [8]. A complete genome-based classification system including 11 dsRNA fragments of RV genome was proposed by the Rotavirus Classification Working Group [9,10]. In terrestrial mammals, there are 18G, 21P, 13I, 4R, 6C, 6M, 9A, 8N, 6T, 5E, and 4H genotypes for the genes VP7, VP4, VP6, VP1, VP2, VP3, NSP1, NSP2, NSP3, NSP4, and NSP5, respectively [11]. Previous reports from China have shown that diarrhea in humans can be caused by RVC [12], but there was no report about RVC in pigs in China until now.

Co-infection with more than one porcine intestinal pathogen is common and the clinical manifestations are more serious, contributing to the outbreak of diarrhea [13,14]. Here, we examined the genomic characteristics of porcine intestinal pathogens infection in pigs and the full-length RVC was detected in the Jiangsu province of China. The whole genome of PoRVC strain belonging to G6P6 genotypes was sequenced, a natural recombination breakpoint in VP2 gene has not been previously reported. This study describes the detection and characterization of RVC in Chinese pig herds. This is also a report of mixed infection of RVC and PEDV. These results provide valuable information on rotavirus recombination and evolution.

## 2. Materials and Methods

In 2018, three intestinal tissue samples of newborn piglets (7-day-old) with diarrhea were collected from a pig farm in the Jiangsu province of China. The collected intestinal contents were reconstituted with phosphate buffer saline (PBS) in a ratio of 1:5. After centrifugation at 8000× *g* at 4 °C for 10 min, the tissue suspension was stored at −20 °C for later use. Porcine epidemic diarrhea virus (PEDV) strain LNsy (Accession No. KY007140) [15], TGEV strain AHHF (KX499468.1), RVA RVA/Pig/China/LNCY/2016/G3P [13], and PDCoV NH (Accession No. KU517165) [16] were used as the control. The RNA was extracted using a QIAamp Viral RNA Mini kit. PrimeScript™ One Step RT-PCR Kit Ver.2 (TaKaRa, Dalian, China) was used for RT-PCR amplification. Primers for amplification of RVC genome and partial gene of SADS-CoV [17], TGEV [18], RVA [19], and PDCoV [16] were designed as Appendix A. PCR products were sequenced by TsingKe (Tsingke Biotechnology Co., Ltd., Beijing, China). The sequences of the RVC and PEDV reference strains were obtained from GenBank, listed in Appendix A, and aligned using the Muscle method in MEGA 5 software. A neighboring–joining tree (bootstrap value, 1000 replicates) was obtained with phylogeny analysis in MEGA software. Classification of gene segments of RVCs was performed according to the definition described by Suzuki and Haseby in 2017 [11]. Recombination analyses were completed using RDP4 software [20].

## 3. Results

To determine the causative agent of intestinal tissue sample of pigs with diarrhea occurring on a pig farm in the Jiangsu province of China, total RNA was extracted from intestinal contents and RT-PCR was performed. The results show that the samples were positive for RVC, as seen in Figure 1A. The whole genome of the PoRVC positive sample was amplified using the primers shown in Appendix A. The 11 segments of the PoRVC genome were successfully amplified, as shown in Figure 1B. The accession numbers of 11 viral segments deposited in GenBank are shown in Table 1. The genotypes for each of the VP7-VP4-VP6-VP1-VP2-VP3-NSP1-NSP2-NSP3-NSP4-NSP5 of RVC/Pig/CHN/JS02/2018/G6P6 were assigned as G6-P6-I5-R1-C1-M3-A10-N1-T1-E1-H1 (Table 1). To monitor whether there is mixed infection of other pathogens in the collected sample, the three intestinal tissue samples were further tested by RT-PCR. The results show that the samples were positive for PEDV, but negative for TGEV, PDCoV, SADS-CoV, and PoRVA, as shown in Appendix A.

The sequences of VP1, VP2, VP3, VP4, VP6, and VP7 were analyzed between RVC/Pig/CHN/JS02/2018/G6P6 sequenced in this study and some RVC strains in GenBank. The VP7 gene contained 1011 nucleotides (nt), encoding 336 amino acids (aa). A phylogenetic tree was constructed using RVC/Pig/CHN/JS02/2018/G6P6 VP7 gene with sequence selected G genotypes in GenBank. The results in Table 1 show that the VP7 gene of RVC/Pig/CHN/JS02/2018/G6P6 was most closely related to RVC/Pig-wt/USA/OK-264/2015 (MF522700) which was isolated from USA at 92.4% and belonged to G6 genotype in Figure 2A. The VP4 gene contained 2208 nt, encoding 735 aa. The VP4 gene of RVC/Pig/CHN/JS02/2018/G6P6 was most closely related to RVC/Pig-wt/KOR/2885/2012/G7PX (KJ814475) isolated from South Korea at 88.5% and belonged to P6 genotype as in Figure 2B. The VP6 gene of RVC/Pig/CHN/JS02/2018/G6P6 was most closely related to RVC/Pig-wt/USA/CO76/2012/G6P5 (MG451721) isolated from USA at 93.2% and both belonged to the I5 genotype as seen in Appendix A. The VP1 gene of RVC/Pig/CHN/JS02/2018/G6P6 was most closely related to PoRVC_VP1_VIRES_HeB02_C2 (MK379289) isolated from China at 95.7% and belonged to the R1 genotype as in Appendix A. The VP2 gene of RVC/Pig/CHN/JS02/2018/G6P6 was most closely related to CJ59-32 (LC307108) isolated from Japan at 90.6% and belonged to the C1 genotype (Appendix A). The VP3 gene of RVC/Pig/CHN/JS02/2018/G6P6 was most closely related to RVC/Pig-wt/Ishi-1/2015/G13P4 (LC122626) isolated from Japan at 97.3%% and belonged to the M3 genotype, as seen in Appendix A.

The sequences of NSP1, NSP2, NSP3, NSP4, and NSP5 were analyzed between RVC/Pig/CHN/JS02/2018/G6P6 sequenced in this study and some RVC strains in GenBank. The results in Table 1 show that the NSP1 sequence was an A10 genotype and was most like PoRVC_NSP1_VIRES_HeB02_C2 (MK379284) isolated from China at 91.3% (Appendix A). The NSP2 sequence was an N1 genotype, most like PoRVC_NSP2_VIRES_HeB02_C1 (MK379286) isolated from China at 97.1% (Appendix A). The NSP3 sequence was a T1 genotype, most like RVC/Pig-wt/CAN/A8-158/2014/G6P4 (KY909964) isolated from Canada at 92.0% (Appendix A). The NSP4 sequence was an E1 genotype, most like RVC/Pig-wt/IND/Por-993/2015 (KY783644) isolated from India at 90.3% (Appendix A). The NSP5 sequence was an H1genotype, most like Cowden (X65938) isolated from USA at 94.5% (Appendix A).

In order to further analyze whether gene recombination exists between RVC/Pig/CHN/JS02/2018/G6P6 and some other RVC isolates in GenBank, we used RDP4 software [20]. Breakpoints for potential recombination zones were found at the VP2 gene at nt 2509 to 2748 in alignment in Figure 3, but no recombination was detected in any other viral genes.

## 4. Discussion

A variety of animal viruses can cause diarrhea in newborn piglets, including PEDV, TGEV, SADS-CoV, PDCoV, and Rotavirus. Rotaviruses cause dehydrating diarrhea in children and young animals [21]. Traditionally, RVA is considered to be the main group of RVs causing diarrhea in suckling piglets, but recent studies have shown that RVC is an endemic disease all over the world and its economic importance cannot be ignored. Further information on the diversity of rotavirus is required to improve understanding of biological characteristics and evolution of rotavirus in China. In 2018, intestinal tissues were collected from a pig farm in Jiangsu province with an outbreak of vomiting and diarrhea. The complete genome of RVC/Pig/CHN/JS02/2018/G6P6 was sequenced and a recombination was identified.

The first detection of RVC was in a piglet with diarrhea in Ohio in 1980 and it was subsequently found in cattle, dogs, ferrets, mink, and humans [6,11]. In recent years, the number of porcine RVC genomes has increased significantly, indicating that it has higher genetic diversity than other host species [22]. The positive rate for PoRVC was detected using a TaqMan Probe-Based Multiplex Real-Time PCR, indicating that the prevalence rate of PoRVC had been previously underestimated [23]. To facilitate the classification of rotavirus strains, a Rotavirus Classification Working Group (RCWG) was formed to set diagnostic guidelines and propose a classification system [24]. By reason of high genetic diversity of rotavirus strains, G (VP7) and P (VP4) genotypes dual typing system was extended to a whole genome sequence classification system [9]. Capsid proteins VP7 and VP4 induced neutralizing antibodies and formed the basis of G and P dual typing system [1]. Previous studies have shown that G6 genotypes were the main G-genotype of RVC [25,26]. This study reported the complete genome of G6P6 subtype RVC in China and attempted to isolate an RVC strain from MA104 and Marc145 cells for virus isolation, ultimately unsuccessfully (data not shown). This may be the reason that RVC strains are very difficult to grow in cell culture [27].

Recombination has been shown to be an important way of virus evolution and a potential mechanism of antigen diversity [28]. In this study, a breakpoint in VP2 gene of RVC was identified, the recombination events are thought to reflect the real situation, contributing to the generation of novel genotypes. On the other hand, only one isolate was sequenced and analyzed in this study, leaving us unable to confirm that genetic recombination has actually occurred in China. Hence, it is necessary to conduct further studies on the recombination of the latest circulating PoRVC strains in China.

Co-infection of multiple pathogens, including PEDV, makes it difficult to control the occurrence of diarrhea in pig farms and have been reported in previous studies. In this study, co-infections of PEDV and PoRVC were identified in China and there was no direct experimental evidence that mixed infection exacerbates diarrhea in pigs. Our findings suggest that PEDV infection may sometimes co-infect with RVC, providing a new perspective for epidemiological investigation of RVC. In this study, PoRVC was detected in one of three samples, while PEDV was detected in three samples. The clinical and pathological relevance of co-infection was not determined because of a small sample size and the lack of epidemiological data. More clinical samples should be collected to detect the co-infection of RVC and PEDV in future, and further studies are still needed to determine the impact of co-infection on virus pathogenicity.

## 5. Conclusions

In summary, the full-length genome of G6P6 genotype of RVC was firstly reported in China, and it was also a case of co-infection with RVC and PEDV. It is worth noting that a new branch A10 formed in the phylogenetic tree analysis of nsp1, combined the recombination events in VP2 of RVC, making the prevention of the swine enteric disease more complicated. These results provide valuable information on rotavirus recombination and will facilitate future investigation about the evolution of the rotavirus.

## Figures and Tables

**Figure 1 animals-12-02951-f001:**
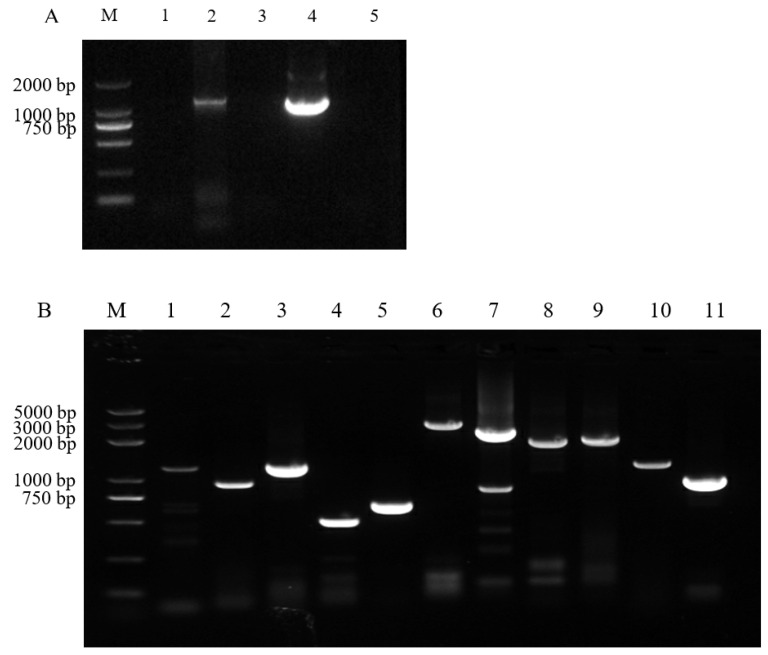
Characterization of RVC infection in collected pig samples. (**A**) RVC PCR amplification of collected samples. M: DL marker 2000; lanes 1–3: collected samples; lane 4: RVC VP6 plasmid; lane 5: negative control. (**B**) Whole-genome amplification of RVC/Pig/China/JS02/2018/G6P6. M: DL2000 plus; lane 1: NSP1; lane 2: NSP2; lane 3: NSP3; lane 4: NSP4; lane 5: NSP5; lane 6: VP1; lane 7: VP2; lane 8: VP3; lane 9: VP4; lane 10: VP6; lane 11: VP7.

**Figure 2 animals-12-02951-f002:**
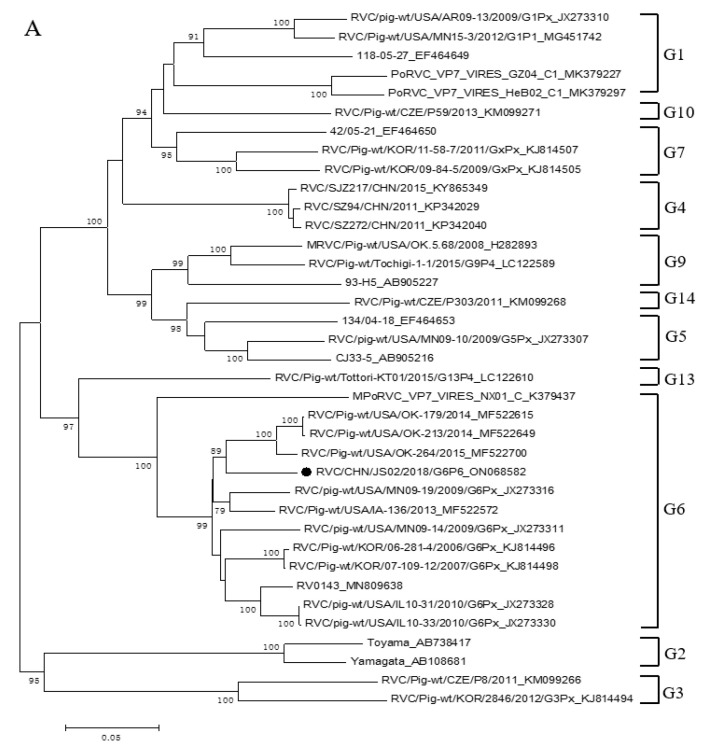
Phylogenetic analysis of RVC/Pig/China/JS02/2018/G6P6. (**A**) VP7 gene; (**B**) VP4 gene.

**Figure 3 animals-12-02951-f003:**
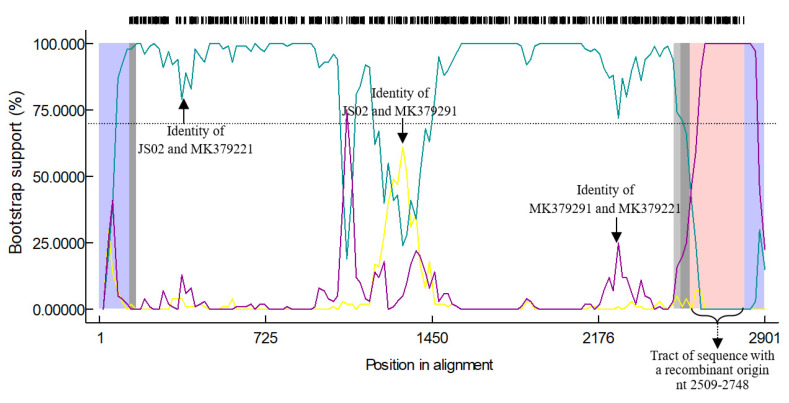
Recombination analysis of RVC/Pig/CHN/JS02/2018/G6P6 VP2 gene with other RVC strains.

**Table 1 animals-12-02951-t001:** Nucleotide and amino acid identities of genome segments of strains RVC/Pig/CHN/JS02/2018/G6P6 compared with the closest strains from GenBank database.

Gene	Closely Related Strains	Nucleotide (%)	Amino Acid (%)	Genotype	Deposited No.
VP1	PoRVC_VP1_VIRES_HeB02_C2(K379289)	95.7%	98.2%	R1	ON068577
VP2	CJ59-32(LC307108)	90.6%	97.6%	C1	ON068578
VP3	RVC/Pig-wt/Ishi-1/2015/G13P [4](LC122626)	97.3%	98.3%	M3	ON068579
VP4	RVC/Pig-wt/KOR/2885/2012/G7PX(KJ814475)	88.5%	92.1%	P6	ON068580
VP6	RVC/Pig-wt/USA/CO76/2012/G6P5(MG451721)	93.2%	98.5%	I5	ON068581
VP7	RVC/Pig-wt/USA/OK-264/2015(MF522700)	92.4%	96.4%	G6	ON068582
NSP1	PoRVC_NSP1_VIRES_HeB02_C2(MK379284)	91.3%	94.4%	A10	ON068572
NSP2	PoRVC_NSP2_VIRES_HeB02_C1(MK379286)	97.1%	99.0%	N1	ON068573
NSP3	RVC/Pig-wt/CAN/A8-158/2014/G6P4(KY909964)	92.0%	95.0%	T1	ON068574
NSP4	RVC/Pig-wt/IND/Por-993/2015(KY783644)	90.3%	84.8%	E1	ON068575
NSP5	Cowden (X65938)	94.5%	95.7%	H1	ON068576

## Data Availability

All data generated or analyzed during this study are included in this article.

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
