# Peer review of "Genome Analysis of the G6P6 Genotype of Porcine Group C Rotavirus in China"

_animals, 2022, doi:10.3390/ani12212951_

Round 1

Reviewer 1 Report (Previous Reviewer 1)

There are some suggestions this reviewer believe can improve the manuscript before publication.

Line 16: PEDV should be spelled out.

Line 51-59. Introduction for coronavirus/PEDV can be redundant and removed, given this research focus on rotavirus sequence. However, I agree it’s good and important to state co-infection of RVC and PEDV in line 65.

Line 65. Extensive language editing is needed in this version, eg. Mixed infection “between” RVC and PEDV should be mixed infection “of” RVC and PEDV.

Results:

There are too many phylogenetic trees (figures)- only one or two representative ones are needed. The corresponding text should be modified.

Do authors use different methods to construct phylogenetic tree and the result is consistent?

Do authors use proof-reading tag for PCR sequencing?

Lines 181-191: this part can be another subsection as PEDV sequence, or moved to beginning under “Co-infection of PEDV and PoRVC was detected in fecal samples”, or simply deleted. Too much details PEDV information can lead out of focus of this rotavirus project.

Discussion:

Line 229. Co-infection of multiple pathogens, “including” PEDV, - not pathogens “with” PEDV.

The scientific writing still can be improved by making the sentences more condensed and easier to be read.

Reference: A current study detects RVC in China should be cited. https://www.mdpi.com/1999-4915/14/8/1819

Author Response

Reviewer 1

There are some suggestions this reviewer believe can improve the manuscript before publication.

  1. Line 16: PEDV should be spelled out.

Response: PEDV has been spelled out in the revised manuscript.

  1. Line 51-59. Introduction for coronavirus/PEDV can be redundant and removed, given this research focus on rotavirus sequence. However, I agree it’s good and important to state co-infection of RVC and PEDV in line 65.

Response: Introduction for coronavirus/PEDV in line 51-59 (previous version) has been removed in the revised manuscript.

  1. Line 65. Extensive language editing is needed in this version, eg. Mixed infection “between” RVC and PEDV should be mixed infection “of” RVC and PEDV.

Response: “Mixed infection “between” RVC and PEDV” has been changed as “Mixed infection of RVC and PEDV” in the revised manuscript.

  1. Results:

There are too many phylogenetic trees (figures)- only one or two representative ones are needed. The corresponding text should be modified.

Response: Some phylogenetic trees have been removed from the revised manuscriopt.

  1. Do authors use different methods to construct phylogenetic tree and the result is consistent?

Response: Thank for reviewer′s suggestions. “A neighboring-joining tree (the bootstrap value was set as 1000 replicates) was obtained with phylogeny analysis, a function of the MEGA software” has been corrected in the revised manuscript.

According to reviewer′s suggestions, two different methods (maximum likelihood and neighboring-joining) were used to construct phylogenetic tree of RVC VP7 gene, the result showed the VP7 gene of RVC/Pig/CHN/JS02/2018/G6P6 sequenced in this study both belonged to G6 genotype. We think that the result is consistent with different methods to construct phylogenetic tree.

  1. Do authors use proof-reading tag for PCR sequencing?

Response: Proof-reading tag for PCR sequencing was not used in this study.

  1. Lines 181-191: this part can be another subsection as PEDV sequence, or moved to beginning under “Co-infection of PEDV and PoRVC was detected in fecal samples”, or simply deleted. Too much details PEDV information can lead out of focus of this rotavirus project.

Response: According to reviewer’s suggestion, “To monitor whether there is mixed infection of other pathogens in collected sample, the three intestinal tissue samples were further tested by RT-PCR. The results showed that the samples were positive for PEDV, but negative for TGEV, PDCoV, SADS-CoV and PoRVA as shown in Figure S1A-E” has been moved to beginning under “Co-infection of PEDV and PoRVC was detected in fecal samples”. Figure S1F and Figure S1G (Some details PEDV information) were deleted in the revised manuscript.

  1. Discussion:

Line 229. Co-infection of multiple pathogens, “including” PEDV, - not pathogens “with” PEDV.

Response: The “with PEDV” has been changed as “including PEDV” in the revised manuscript.

  1. The scientific writing still can be improved by making the sentences more condensed and easier to be read.

Response: We have revised the manuscript according to the suggestion, which has greatly improved the level of our manuscript. Thank you for the suggestion.

  1. Reference: A current study detects RVC in China should be cited. https://www.mdpi.com/1999-4915/14/8/1819

Response: The reference “A TaqMan Probe-Based Multiplex Real-Time PCR for Simultaneous Detection of Porcine Epidemic Diarrhea Virus Subtypes G1 and G2, and Porcine Rotavirus Groups A and C” has been added in the revised manuscript.

Reviewer 2 Report (Previous Reviewer 3)

Congratulations for this study, is interesting

Author Response

Reviewer 2

Congratulations for this study, is interesting

Response: Thanks for reviewer’s suggestion.

Reviewer 3 Report (New Reviewer)

Please check the attached pdf file in which comments and questions are added.

Author Response

Reviewer 3

  1. "biological" is a word with broad significance. I would have used word such as " genomic"

Response: Changed as reviewer’s suggestion.

  1. where in China? You have limited number of samples (3) from specific farm.

Response: “Jiangsu province of China” has been added in the revised manuscript.

  1. you just collected 3 samples. Are these samples from a farm or different farms?

Response: The “a pig farm” has been added in the revised manuscript.

  1. use the word "reconstituted with" rather than "mixed with"

Response: Changed as reviewer’s suggestion.

  1. do you mean the "pellet" ? there are a lot of stuff in the intestinal content not only intestinal tissues

Response: Changed as reviewer’s suggestion.

  1. The gel picture shows the whole 3 tested samples are positive. Why is it 33 %? how did you measure the percentage?

Response: The mistake has been corrected in the revised manuscript.

  1. what does this acronym stands for?

Response: JSCF has been deleted in the revised manuscript.

  1. do you mean among the three samples or isolates you collected? Do they look alike or genetically identical ?

Response: The mistake has been changed as “sequenced in this study and some RVC strains in Gen-Bank” in the revised manuscript.

  1. please give reference

Response: The references (Saif et al., 1980; Suzuki and Hasebe, 2017)” have been added in the revised manuscript.

Reviewer 4 Report (New Reviewer)

In this manuscript, authors claimed that they identified a porcine rotavirus group C from one of the three swine intestinal samples from pig farm in China and found a natural recombination event occurred in VP2 region of the virus genome. My opinion is stated as follows:

1.     Lines 18-19: The sentence “A natural recombination event was observed and the breakpoint at nucleotides (nt) 2,509 to 2,748 in VP2 gene” can be rewritten as “A natural recombination event was observed with breakpoints at nucleotides (nt) 2,509 to 2,748 of the VP2 gene”.

2.     Line 26: Natural's N should be lowercase.

3.     Line 27: The data presented here is not related to the prevalence of RVC that exists in China. Only one case should not use the term "epidemic".

4.     Line 65: The sentence “This was also a report of mixed infection between RVC and PEDV” can be rewritten as “This was also a report of a mixed infection of RVC and PEDV”.

5.     Lines 98-99: Authors stated that only three intestine samples were used in this study. I guessed lane 1 to lane 3 in Figure 1A were the three different intestinal samples, so what does this sentence meant "the samples were 33% positive for RVC as seen in Figure 1A".

6.      Lines 101-103: The sentence can be rewritten as “The accession numbers of 11 viral segments deposited in GenBank are shown in Table 1”.

7.      Lines 168-172: This paragraph should be rewritten.

8.      Figure 3: Where are Figures 3A, 3B, and 3C?

9.      Lines 216-217: It’s for virus isolation, not for cell isolation.

10.   Lines 218-220: This sentence should be rewritten.

11.   Line 241: … RVC was firstly reported in China.

12.   Only one isolate was sequenced and analyzed in this study unable to confirm that genetic recombination has actually occurred in China, it is recommended to isolate more sequences to confirm this conclusion.

13.  It is recommended to ask native English speakers to help review and revise this article.

Author Response

Reviewer 4

In this manuscript, authors claimed that they identified a porcine rotavirus group C from one of the three swine intestinal samples from pig farm in China and found a natural recombination event occurred in VP2 region of the virus genome. My opinion is stated as follows:

  1. Lines 18-19: The sentence “A natural recombination event was observed and the breakpoint at nucleotides (nt) 2,509 to 2,748 in VP2 gene” can be rewritten as “A natural recombination event was observed with breakpoints at nucleotides (nt) 2,509 to 2,748 of the VP2 gene”.

Response: “A natural recombination event was observed with breakpoints at nucleotides (nt) 2,509 to 2,748 of the VP2 gene” has been added in the revised manuscript.

  1. Line 26: Natural's N should be lowercase.

Response: Revised as reviewer’s suggestion.

  1. Line 27: The data presented here is not related to the prevalence of RVC that exists in China. Only one case should not use the term "epidemic".

Response: The "epidemic of" has been deleted in the revised manuscript.

  1. Line 65: The sentence “This was also a report of mixed infection between RVC and PEDV” can be rewritten as “This was also a report of a mixed infection of RVC and PEDV”.

Response: “This was also a report of a mixed infection of RVC and PEDV” has been added in the revised manuscript.

  1. Lines 98-99: Authors stated that only three intestine samples were used in this study. I guessed lane 1 to lane 3 in Figure 1A were the three different intestinal samples, so what does this sentence meant "the samples were 33% positive for RVC as seen in Figure 1A".

Response: The Figure 1A was changed in the revised manuscript.

  1. Lines 101-103: The sentence can be rewritten as “The accession numbers of 11 viral segments deposited in GenBank are shown in Table 1”.

Response: “The accession numbers of 11 viral segments deposited in GenBank are shown in Table 1” has been added in the revised manuscript.

  1. Lines 168-172: This paragraph should be rewritten.

Response: This paragraph has been rewritten in the revised manuscript.

  1. Figure 3: Where are Figures 3A, 3B, and 3C?

Response: “Figures 3A, 3B, and 3C” has been deleted in the revised manuscript.

  1. Lines 216-217: It’s for virus isolation, not for cell isolation.

Response: Revised as reviewer’s suggestion.

  1. Lines 218-220: This sentence should be rewritten.

Response: This sentence has been deleted in the revised manuscript.

  1. Line 241: … RVC was firstly reported in China.

Response: Revised as reviewer’s suggestion.

  1. Only one isolate was sequenced and analyzed in this study unable to confirm that genetic recombination has actually occurred in China, it is recommended to isolate more sequences to confirm this conclusion.

Response: Thank you for the suggestion. “The ongoing discovery of recombination events indicated that PoRVC infection in Chinese farms is becoming more complicated” has been deleted in the revised manuscript.

“On the other hand, only one isolate was sequenced and analyzed in this study, which unable to confirm that genetic recombination has actually occurred in China” has been added in the revised manuscript.

  1. It is recommended to ask native English speakers to help review and revise this article.

 Response: We have revised the manuscript according to the suggestion, which has greatly improved the level of our manuscript. Thank you for the suggestion.

Reviewer 5 Report (New Reviewer)

Jiao et al.  provided the complete genome sequence of a RCV strain detected in China in this study. A nature recombination event was identified in the vp2 gene. These data will contribute to the ubderstanding of the evolution of rotavirus.

Author Response

Reviewer 5

Jiao et al.  provided the complete genome sequence of a RCV strain detected in China in this study. A nature recombination event was identified in the vp2 gene. These data will contribute to the ubderstanding of the evolution of rotavirus.

Response: Thanks for reviewer’s suggestion.

This manuscript is a resubmission of an earlier submission. The following is a list of the peer review reports and author responses from that submission.

Round 1

Reviewer 1 Report

In this study, only 3 samples were included. Porcine rotavirus C (RVC) genome was detected in one of 3 samples, while porcine epidemic diarrhea virus (PEDV) was detected in 3 samples. Given that PEDV and RVC are common viral pathogens widely spread in conventional pig farms, it is not unusual for these 2 viruses to be detected in the same sample. The clinical and pathological relevance of co-infection was not determined because of a small sample size and lack of epidemiologic data, and was very likely (base on the result of re-inoculation) an incidental finding.

The RT-PCR was conducted by PrimeScript™ One Step RT-103 PCR Kit Ver.2 in the present study. Is it a proofreading PCR for sequencing?

The pathology images can be improved by adjusting the orientation (upside down of villi in this manuscript) and removing the light blue background.

The contents in the Discussion are more about literature review without insight from the study result.

Considering the scientific merit, this reviewer does not suggest a publication of this project at this stage.  Alternatively, the editor may consider suggesting a short/brief communication focus on genomic analysis/ first publication of the full genome of RVC in China.

Reviewer 2 Report

There are many pathogens causing swine diarrhea, especially viruses. The manuscript detected PEDV and RVC of diarrhea samples collected from pig farms, and find one case of co-infection with RVC and PEDV. They also observed a natural recombination event which the breakpoint at nucleotides (nt) 2,509 to 2,748 in VP2 gene of RVC. Due to the high incidence rate of the current swine diarrhea epidemic, and it is usually a mixed infection, these findings would provide a new insight into the evolution of rotavirus and co-infection with swine enteric coronaviruses. This manuscript was scientifically sound and well written; only one minor revision is required as below.

1. References 5, 6, 15, 19, 56, and 68, missing page number.

Reviewer 3 Report

The manuscript is well written and has the appropriate methodology, but I don't agree it's good practice to state it is "first paper to publish these findings. The title gives too much relevance to being the "first" and takes away relevance to other findings